# Prevalence, patterns and correlates of smokeless tobacco use in Nigerian adults: An analysis of the Global Adult Tobacco Survey

Ikenna Onoh[1]*, Oluwatomi Owopetu[2], Abdulhakeem Abayomi Olorukooba[3], Chukwuma David Umeokonkwo[4], Tukur Dahiru[3], Muhammad Shakir Balogun[5]

1 Nigeria Field Epidemiology and Laboratory Training Programme, Abuja, Nigeria, 2 Department of Community Medicine, University College Hospital, Ibadan, Nigeria, 3 Department of Community Medicine, Faculty of Clinical Sciences, College of Medical Sciences, Ahmadu Bello University, Zaria, Kaduna State, Nigeria, 4 Department of Community Medicine, Alex Ekwueme Federal University Teaching Hospital, Abakaliki, Ebonyi State, Nigeria, 5 African Field Epidemiology Network, Abuja, Nigeria

* ikeonoh@yahoo.com

**Data Availability Statement:** The data underlying the results presented in this study are available from the Centers for Disease Control and

## Abstract

### Introduction

The global tobacco epidemic contributes to more than 8 million deaths annually. However, most tobacco control interventions have been driven by an emphasis on smoked tobacco. Globally and more so in Nigeria, less attention has been paid to the similarly harmful smokeless tobacco (SLT) whose use appeals to a different demography. We examined the prevalence, patterns of use and correlates of SLT in Nigerian adults to guide targeted control efforts.

### Methods

We conducted a secondary analysis of the 2012 Global Adult Tobacco Survey (GATS) data. We obtained data on 9,765 non-institutionalised adults aged 15 years and older. Variables included current SLT use, sociodemographic characteristics and perceived harm of SLT use. We used Chi-square test to examine associations and binary logistic regression to assess predictors of current SLT use. All analyses were conducted with sample-weighted data.

### Results

The prevalence of current SLT use was 1.9% of all adults. About 1.4% were daily users. The main types were snuff by nose (1.6%) and snuff by mouth (0.8%). There were higher odds of current SLT use for those in the South-East region (aOR = 13.99; 95% CI: 4.45–43.95), rural area residents (aOR = 1.56; 95% CI: 1.04–2.35), males (aOR = 4.43; 95% CI: 2.75–7.11), the 45–64 years age-group (aOR = 10.00; 95% CI: 4.12–24.29), those with no formal education (aOR = 2.67; 95% CI: 1.01–7.05), and those with no perception of harm from SLT use (aOR = 3.81, 95% CI: 2.61–5.56).

Prevention at this URL: https://nccd.cdc.gov/
GTSSDataSurveyResources/Ancillary/DataReports.
aspx?CAID=2&Survey=4&WHORegion=
3&Country=230&Site=556000.

**Funding:** The author(s) received no specific
funding for this work.

**Competing interests:** The authors have declared
that no competing interests exist.

## Conclusion

The prevalence of SLT use among Nigerian adults was low with clearly identified predictors. While a majority were aware of harm from SLT use, an unacceptably high proportion remain unaware. We recommended targeted interventions to increase awareness of the harmful effects of SLT use especially among residents of the South-East, those in rural areas, males, and individuals with no formal education. We also recommended a follow-up survey.

## Introduction

The association between tobacco use and the risk of development of several health effects are well documented in literature. These consequences are borne by millions globally, contributing significantly to morbidity and mortality for current and future generations [1–4], annually accounting for about 8 million deaths worldwide [5]. Every 6 seconds one person dies due to tobacco-related disease. About 75% of this mortality is among residents of low- and middle-income countries [1]. The World Health Organization (WHO) responded to this epidemic by developing the Framework Convention on Tobacco Control [6], which has undergone varying levels of implementation in different countries. However, most of these efforts aimed at tobacco control interventions have been driven by an emphasis on smoked tobacco particularly manufactured cigarettes [7], as its prevalence of use is about three times that of SLT [8]. Less attention has been paid to the similarly harmful smokeless tobacco (SLT), whose serious health effects have been documented [2, 9, 10].

SLT use adversely impacts health, with outcomes such as periodontal disease, mouth lesions and leukoplakia [9, 10]. Also, a 2012 International Agency for Research on Cancer (IARC) review of multiple studies showed that SLT causes oral cancer, esophageal cancer, and pancreatic cancer in humans [11]. Furthermore, SLT users upon cessation of use may show signs of nicotine addiction, such as tolerance and withdrawal symptoms [12].

Although SLT use causes adverse health outcomes, understanding its use and the impact is made difficult by the wide variation in products and user-specific behaviors. Globally, a wide range of SLT products with varying characteristics are available including snus, snuff, chewing tobacco, gutka, betel quid with tobacco, toombak, iqmik and tobacco lozenges [9]. This diversity makes generalizing about these products across regions and countries as one entity inaccurate [9]. Furthermore, SLT product regulation (e.g. via taxes/marketing restrictions), production, and sales vary widely across countries and regions [9].

Recent estimates show global prevalence of current SLT use to be 5.6% [8]. South East Asia bears a disproportionately high burden with a prevalence of 17.9% [8]. African prevalence stands at 2.8% [8]. Some of the previously observed patterns of use include predominant daily use [9, 13], and dual-use corresponding to underlying SLT and smoked tobacco prevalences [9, 14]. Some of the factors that have been found to be associated with SLT use include increasing age, male gender, lower educational attainment and lack of awareness of consequent health risks [7, 10, 13–17].

Evidence on SLT use prevalence and its associated factors in Nigeria is sparse. A recent national survey, the 2018 Nigeria Demographic and Health Survey (NDHS) revealed female and male prevalences of 0.1% and 1.2% respectively [18]. But the NDHS methodology only takes a cursory look at tobacco use and associated factors in a narrower age band compared to the Global Adult Tobacco Survey (GATS). A localised isolated study in Northern Nigeria showed a prevalence of 7.49% [7]. The need to study SLT use in Nigeria becomes more

important as there is an anticipated increased prevalence from addicted smokers seeking alternative nicotine sources as tobacco control efforts emphasizing on smoked tobacco [19, 20] become intensified and more effective.

In 2012, the only Global Adult Tobacco Survey (GATS) in Nigeria till date was carried out to "establish baseline information on tobacco use and tobacco control measures in a nationally representative sample with regard to exposure to secondhand smoke, cessation, risk perceptions, knowledge and attitudes, exposure to media, price and taxation issues by using a global standard protocol adapted to country-specific context" [1]. We analyzed smokeless tobacco data from this survey to determine the prevalence, patterns of use and underlying correlates in Nigerian adults to guide targeted control efforts. With the passing of the Nigeria Tobacco Control Act 2015, findings from the analysis will provide baseline country-specific information for future evaluation of the impact of the legislation in the control of tobacco use. It will also provide valuable information to enable appropriate emphasis on SLT in the development of guidelines and other regulations critical for the control of tobacco use in Nigeria.

## Materials and methods

### Study area

Nigeria is located in West Africa between latitudes 4˚16'N and 13˚53'N North and longitudes 2˚40'E and 14˚41'E with a land area of 923,768 square kilometres. It has Africa's largest population, estimated at 214,028,302 [21]. Nigeria is Africa's largest economy, with 2017 GDP (purchasing power parity) estimated at US$1.121 trillion [21]. Agriculture is a significant sector of the economy especially as regards engagement of labour. Cash crops including tobacco account for a significant part of agricultural activities. Nigeria is home to transnational tobacco companies, including British America Tobacco Nigeria and Japan Tobacco International. Historically, the SLT products available locally have been unbranded and unpackaged cottage samples made by sun-drying or fire-curing, branded sources have been negligible [22, 23].

The history of anti-tobacco legislation started in 1958 with the Tobacco Ordinance [20], followed in 1990 by the Tobacco Smoking Control Decree 20 [1]. The WHO Framework Convention on Tobacco Control (FCTC) was signed and ratified in 2004 and 2005 respectively [1]. The National Tobacco Control Act (the NTCA) [19], a law to domesticate the WHO FCTC was enacted in 2015.

### Data sources

This study used data from the 2012 GATS conducted in Nigeria [1], the only one done in the country till date. GATS is the flagship survey for adult tobacco use and control surveillance [24]. In Nigeria, the implementing agency for GATS was the National Bureau of Statistics (NBS) with oversight from the Federal Ministry of Health (FMOH). The World Health Organization (WHO) and the U.S. Centers for Disease Control and Prevention (CDC) provided technical support. The Bloomberg Initiative to Reduce Tobacco Use provided funding [1].

The survey design, using a multistage stratified cluster sampling technique, enables the generation of precise national level estimates, by geographical locality (urban/rural) and gender. It also allows for the comparison of estimates among the six regions of Nigeria [1]. Within each state, a sample of enumeration areas (EAs) were systematically selected from a master sample, with probability proportional size (PPS). These EAs constituted the primary sampling units (PSU) for the survey. A systematic sample of households (secondary sampling units, SSU) were selected from each EA, and then an eligible individual from each selected household. A total of 1,100 EAs (300 urban and 800 rural) were selected, and from these 11,1107 households (5,776 urban and 5,331 rural) [1]. The study population for this household survey was non-

institutionalised individuals 15 years of age and older. The data were collected at two levels, the household and individual levels. Out of the 11,107 households sampled, screening was completed by 9,911. Interviews were completed for 9,765 individuals (one individual randomly chosen from each selected household). The overall, household and individual response rates were 89.1%, 90.3% and 98.6% respectively [1].

## Variables

**Outcome variable.** Current smokeless tobacco use was assessed using respondents' self-report of current daily, less than daily or none use of smokeless tobacco. Daily and less than daily (occasional) current users were classified as current smokeless tobacco users. Dual users were current users of both smokeless and smoked tobacco.

**Independent variables.** The socio-demographic variables included sex, age, urban/rural residence, region, level of education, and employment status. Some of the variables were re-coded to attain reasonable sample sizes, and also based on suggestions in the literature. Age in years was recoded into 4 groups, "15–24", "25–44", "45–64", and "65+" years. With regards to the level of education, "no formal education" was unaltered, "less than primary completed" and "primary completed" were recoded as "primary", "junior secondary completed" and "senior secondary completed" were recoded as "secondary", "less than college/university completed" and "college/university completed" were recoded as "post-secondary". For employment status, "self-employed", "student", and "home maker" were unaltered, while "government employee" and "non-government employee" were recoded to "employee", and "retired", "unemployed able to work", "unemployed unable to work" were grouped together.

Perceived risk of smokeless tobacco use was assessed using a question that asked, "Based on what you know or believe, does using smokeless tobacco cause serious illness?" The possible responses were "yes", "no", and "don't know". "No" and 'don't know" were recoded together for this study.

## Data analysis

We used SPSS version 23 to perform the data analysis. Frequencies and proportions were generated to describe the distribution of the independent and outcome variables. We used the Chi-Square test to assess bivariate associations between independent variables and current smokeless tobacco use. A binary logistic regression model was also developed for predictors of current smokeless tobacco use. The enter variable selection method was used in building the regression model. All statistical tests were set at 5% level of significance, and all analyses were conducted with sample-weighted data.

## Ethical considerations

This study was a secondary analysis of an open source dataset which was downloaded from the Global Tobacco Surveillance System Data (GTSS Data) [25], a web-based application domiciled within the CDC website, that houses and displays data from four tobacco-related surveys conducted around the world. The survey data sets used in this study were completely anonymous. The GATS Nigeria protocols were approved by the CDC. During the survey, data collectors obtained written informed consent of participants. In addition, written informed parental consent was obtained where participant was 17 years old or less. Interviewers ensured the confidentiality of the data they had collected and signed the GATS statement on confidentiality [1].

## Results

### Participation rates and socio-demographic characteristics

A total of 9765 respondents were studied. Their basic characteristics are as shown in Table 1.

### Prevalence and patterns of current smokeless tobacco use

The prevalence of smokeless tobacco use among Nigerians aged 15 years and above was 1.9% (95% CI: 1.6%-2.3%). The sex-specific analysis shows a prevalence of 3.0% (95% CI: 2.4%-3.7%) in males and 0.9% (95% CI: 0.7%-1.2%) in females. About three-quarters of this 1.9%, i.e., 1.4% (95% CI: 1.1%-1.8%) of the total adult population were daily users while the rest use smokeless tobacco occasionally. About 0.2% (95% CI: 0.1%-0.4%) of the whole population use both smoked and smokeless tobacco. Furthermore, Fig 1 shows the various types of smokeless tobacco currently used by adults in Nigeria. The highest was snuff by nose, followed by snuff by mouth and other types comprising of drinking and chewing tobacco.

**Table 1. Socio-demographic characteristics of Nigerian adults.**

| Variables | (N = 9765) | |
|---|---|---|
| | % | 95% C.I. |
| **Age group (years)** | | |
| 15–24 | 34.4 | 32.9, 35.9 |
| 25–44 | 43.8 | 42.3, 45.2 |
| 45–64 | 16.2 | 15.2, 17.3 |
| 65+ | 5.6 | 5.0, 6.2 |
| **Sex** | | |
| Male | 50.0 | 48.5, 51.6 |
| Female | 50.0 | 48.4, 51.5 |
| **Residence** | | |
| Urban | 37.0 | 33.9, 40.1 |
| Rural | 63.0 | 59.9, 66.1 |
| **Level of education* (n = 9753)** | | |
| No formal education | 31.0 | 29.2, 32.8 |
| Primary or less | 19.2 | 18.1, 20.5 |
| Secondary | 38.0 | 36.4, 39.7 |
| Post-secondary | 11.8 | 10.7, 13.0 |
| **Employment status* (n = 9756)** | | |
| Employee | 10.7 | 9.8, 11.8 |
| Self-employed | 49.4 | 47.7, 51.1 |
| Student | 18.7 | 17.4, 20.1 |
| Home maker | 14.0 | 13.0, 15.1 |
| Retired/Unemployed | 7.2 | 6.3, 8.0 |
| **Region** | | |
| North-Central | 13.7 | 12.6, 14.9 |
| North-East | 12.4 | 11.3, 13.6 |
| North-West | 23.1 | 21.6, 24.7 |
| South-East | 12.8 | 11.5, 14.1 |
| South-South | 16.2 | 14.9, 17.7 |
| South-West | 21.8 | 20.3, 23.3 |

*Variables with missing data

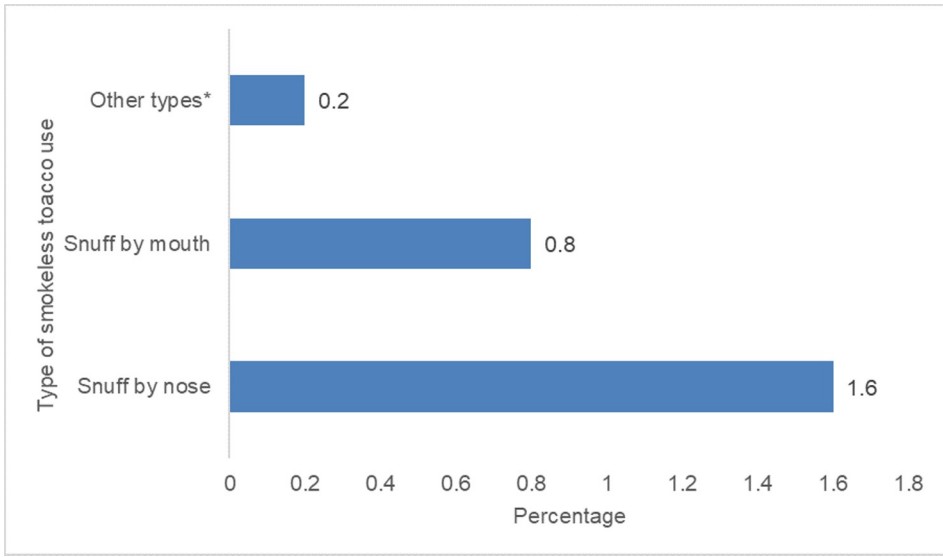

**Fig 1. Prevalence of types of current smokeless use in the population**†. * Chewing and drinking tobacco. † These categories are not mutually exclusive.

## Perception of risk of smokeless tobacco use

With regards to the population's perception of the harmfulness of smokeless tobacco use, 31.1% (95% CI: 29.4%-32.9%) of Nigerian adults were either unaware or perceived smokeless tobacco to be harmless.

## Determinants of current smokeless tobacco use

Table 2 shows the bivariate associations between selected independent factors and current smokeless tobacco use in Nigerian adults 15 years and above. All the variables were found to be statistically significant. Higher age groups above 64 years (8.3%), males (3.0%), rural dwellers (2.3%), those with no formal education (2.8%), self-employed (2.9%) and from the south-east region (4.7%) had the highest relative prevalence of use.

Table 3 shows the predictors of smokeless tobacco use in Nigerian adults. Significant predictors of use include age group, sex, place of residence, region and risk perception. Those 45–64 years had 10 times (OR = 10.00; 95% CI: 4.12–24.29) higher odds of SLT use compared to those 15–24 years, those 65 years and above had similar odds (OR = 9.70; 95% CI: 3.51–26.79). Males were found to have about 4 times (OR = 4.43; 95% CI: 2.75–7.11) higher odds than females. Rural dwellers had about 1.5 times (OR = 1.56; 95% CI: 1.04–2.35) higher odds than urban dwellers. The south-east region had about 14 times (OR = 13.99; 95% CI: 4.45–43.95) higher odds followed by the north-east with similar odds (OR = 13.63; 95% CI: 4.75–39.16). Those who were either unaware or perceived SLT not to be harmful had about 4 times (OR = 3.81; 95% CI: 2.61–5.56) higher odds relative to those who perceived it as harmful.

## Discussion

This study examined the prevalence, patterns and underlying risk factors for smokeless tobacco use among non-institutionalised Nigerian adults. The findings show a relatively low prevalence. The commonest form of use was snuff by nose, with a majority as daily users. Proper perception of risk of use was low and some of the determinants of use included low

**Table 2. Association between selected characteristics and current smokeless tobacco use in Nigerian adults.**

| Variables (N = 9747) | Current Smokeless tobacco use status (%) | | Crude Odds Ratio (cOR) | cOR 95% CI | P-value† |
|---|---|---|---|---|---|
| | User | Non-user | | | |
| **Age group (years)** | | | | | <0.001 |
| 65+ | 8.3 | 91.7 | 39.29 | 13.44–114.87 | |
| 45–64 | 5.1 | 94.9 | 23.46 | 8.04–68.48 | |
| 25–44 | 1.3 | 98.7 | 5.48 | 1.86–16.09 | |
| 15–24 | 0.2 | 99.8 | 1 | - | |
| **Sex** | | | | | <0.001 |
| Male | 3.0 | 97.0 | 3.38 | 2.31–4.93 | |
| Female | 0.9 | 99.1 | 1 | - | |
| **Residence** | | | | | 0.001 |
| Rural | 1.3 | 98.7 | 1.85 | 1.27–2.71 | |
| Urban | 2.3 | 97.7 | 1 | - | |
| **Level of education* (n = 9735)** | | | | | <0.001 |
| No formal education | 2.8 | 97.2 | 3.13 | 1.47–6.65 | |
| Primary or less | 2.6 | 97.4 | 2,83 | 1.33–6.02 | |
| Secondary | 1.1 | 98.9 | 1.24 | 0.57–2.70 | |
| Post-secondary | 0.9 | 99.1 | 1 | - | |
| **Employment status* (n = 9738)** | | | | | <0.001 |
| Employee | 1.3 | 98.7 | 3.77 | 1.01–13.99 | |
| Self-employed | 2.9 | 97.1 | 8.27 | 2.57–26.57 | |
| Home maker | 0.8 | 99.2 | 2.13 | 0.57–7.99 | |
| Retired/Unemployed | 2.5 | 97.5 | 7.20 | 2.09–24.82 | |
| Student | 0.4 | 99.6 | 1 | - | |
| **Region** | | | | | <0.001 |
| South-East | 4.7 | 95.3 | 23.51 | 8.89–62.15 | |
| North-East | 3.0 | 97.0 | 14.54 | 4.81–43.95 | |
| North-Central | 3.2 | 96.8 | 16.02 | 5.86–43.82 | |
| South-West | 1.3 | 98.7 | 6.22 | 2.28–16.96 | |
| South-South | 1.2 | 98.8 | 5.74 | 2.02–16.35 | |
| North-West | 0.2 | 99.8 | 1 | - | |
| **Risk perception* (n = 9736)** | | | | | <0.001 |
| Not harmful/Don't know | 3.9 | 96.1 | 4.01 | 2.81–5.73 | |
| Harmful | 1.0 | 99.0 | 1 | - | |

*Variables with missing data.

†P-value for the Chi-square test

risk perception, male sex, increasing age, rural residence, no formal education and residence in the north- and south-eastern regions of the country.

The overall prevalence of SLT use in the study population appeared low in comparison to recent global and African averages [8], and a previous study in a city in the north-east of Nigeria [7], similar to figures from a secondary analysis of data from the 2013 Nigeria Demographic and Health Survey (NDHS) [2], and higher than figures from the 2018 NDHS [18]. The low figures in comparison to the global average is due to the high prevalence rates in the South-East Asian region which bears about 80% of SLT use burden and where nearly 20% of the population use SLT [8]. Analysis of large population-based surveys across Africa have also shown

**Table 3. Predictors of current smokeless tobacco use amongst Nigerian adults.**

| Variables | Adjusted Odds Ratio (aOR) | aOR 95% CI | P-value |
|---|---|---|---|
| **Age group (years)** | | | |
| 65+ | 9.70 | 3.51–26.79* | <0.001 |
| 45–64 | 10.00 | 4.12–24.29* | <0.001 |
| 25–44 | 3.41 | 1.43–8.09* | 0.006 |
| 15–24 | 1 | - | - |
| **Sex** | | | |
| Male | 4.43 | 2.75–7.11* | <0.001 |
| Female | 1 | - | - |
| **Residence** | | | |
| Rural | 1.56 | 1.04–2.35 | 0.034 |
| Urban | 1 | - | - |
| **Level of education** | | | |
| No formal education | 2.67 | 1.01–7.05 | 0.047 |
| Primary or less | 1.84 | 0.76–4.46 | 0.175 |
| Secondary | 1.69 | 0.73–3.92 | 0.219 |
| Post-secondary | 1 | - | - |
| **Employment status** | | | |
| Employee | 0.29 | 0.07–1.21 | 0.088 |
| Self-employed | 0.51 | 0.14–1.83 | 0.300 |
| Home maker | 0.43 | 0.10–1.82 | 0.248 |
| Retired/Unemployed | 0.60 | 0.15–2.37 | 0.467 |
| Student | 1 | - | - |
| **Region** | | | |
| South-East | 13.99 | 4.45–43.95* | <0.001 |
| North-East | 13.63 | 4.75–39.16* | <0.001 |
| North-Central | 11.52 | 3.76–35.32* | <0.001 |
| South-West | 7.24 | 2.35–22.32* | 0.001 |
| South-South | 3.82 | 1.12–13.04* | 0.032 |
| North-West | 1 | - | - |
| **Risk perception** | | | |
| Not harmful/Don't know | 3.81 | 2.61–5.56* | <0.001 |
| Harmful | 1 | - | - |

*Statistically significant

a low prevalence of SLT use among men and women in West Africa as compared to other regions [26], findings probably accounted for by varying cultural and social norms. The north-eastern region of Nigeria has consistently shown a higher level of SLT use as revealed in the index study as well, this due to a high level of social acceptance of SLT [7]. The NDHS findings in combination with ours, seem to suggest a downward trend, but the differing methodologies, and just 2 time points preclude the validity of this conclusion.

Most of the SLT users in our study were daily users. This is similar to findings in Pakistan [13], as well as in Bangladesh, India and Myanmar [9]. This relative preponderance of daily users is likely due to the well-established addiction initiating and perpetuating effects of nicotine, a notable challenge to tobacco cessation interventions. SLT users are known to crave and continue to use it even when harmful to their health, they sometimes switch to products with higher nicotine levels, and are not able to sustain attempts at cessation of use [27]. In an earlier

study among an elderly Nigerian population half of the respondents were dependent on snuff [28]. A much lower proportion of the aggregate Nigerian population were dual (smoked and smokeless) tobacco users relative to figures from India [14], Bangladesh [9] and north-east Nigeria [7]. This difference is reflective of the already documented higher prevalence of the individual products in these other populations. Dual-use is an important public health metric as dual users when compared with single users, are at a greater risk for morbidity and mortality [29]. Snuff by nose was the predominant SLT type, followed by snuff by mouth. A negligible proportion used other forms such as chewing and drinking tobacco. Other local studies have shown similar patterns [2, 7]. There are two main types of smokeless tobacco: snuff (moist and dry) and chewing tobacco [30], the specific types, formulations and names vary significantly by country and region [9]. The moist snuff predominates in Africa generally [31], but the dry snuff most commonly inhaled through the nose, is historically and culturally local to Nigeria, Cameroon, Senegal and Chad [9].

A significant percentage of the population was unaware of the risk of serious illness due to SLT use. This was much higher than what was found in Uganda amongst a similar population [10] as well as for cigarette use in our study population [1]. Some of the reasons behind this comparative lower levels of awareness include social acceptance [7], regular use of SLT products in social and traditional gatherings such as marriages [9, 31], perceived medicinal properties [7, 9], marketing of SLT as a convenient and harm reducing alternative to cigarette smoking, and less emphasis on SLT use control as compared to cigarette smoking [9]. These factors independently and through the mediating effect of lower levels of awareness of harm may serve as critical drivers of increased SLT use in response to intensified marketing efforts, while also impinging on efforts at getting users to quit. Well-designed, sustained, comprehensive health education efforts and enlightenment campaigns will serve to attenuate the effect of the interplay of these factors.

There were significantly higher odds of current SLT use associated with increasing age, male gender, rural location, having no formal education, having no perception of risk of SLT use and living in all other regions of the country as compared to the north-west. Increasing age has been severally found to be associated with SLT use [7, 13, 16, 17]. With the difficulty of cessation for earlier onset users, newer users are cumulatively added to older cohorts thus increasing the likelihood in older age. It is also thought that there might be a switch from smoked to smokeless tobacco in the later stages of the lifespan in a bid to minimise perceived health risks while retaining the nicotine addiction [32]. Other studies also agree with the higher odds of use in males [7, 13, 16], and this is likely due to their higher risk-taking behaviour and cultural/social acceptance. The higher rural burdens similarly reported previously seem to stem from the fact that SLTs are integral to many social gatherings [31] such as traditional marriages which are domiciled in the rural areas. And even though SLT products available in Africa range from manufactured to home-made, most are home-made and these are more likely to be used by rural dwellers [31]. Our findings are also consistent with a widely documented [7, 14, 15] inverse relationship between educational attainment and SLT use. This relationship may be linked to these individuals' knowledge and therefore perception about the possible harmful effects of SLT use, another significant predictor of use in the study population. Those unaware of these risks were significantly more likely to use SLT, similar to a Ugandan study [10]. Regional differences have been shown in other countries [16, 33], but since this is the first detailed national-level study of SLT use in Nigeria, there are no similar study results to compare it to. These differences are likely due to long-standing cultural differences and social norms.

The current study had some strengths and limitations. With regards to the strengths, the sample size was large enough to ensure the validity of the various analyses and sub-group

analyses. Secondly, these respondents were selected using systematic and rigorous methods that ensured a good representation of the sub-groups represented within the larger Nigerian population. Therefore, these findings are generalisable to the population. The first limitation derives from the cross-sectional study design so that a temporal sequence is not demonstrable between independent variables and the outcome variable. Interpretations of associations should therefore, be cautiously made. Secondly, all the information especially the history of SLT use as obtained during the survey, is self-reported with no measured bio-markers for verification. In the relatively conservative Nigerian context, there could have been under-reporting due to the desire for conformity to social norms. Thirdly, there were missing data for certain key variables, but the proportions in all cases were likely not high enough to introduce any form of bias. Also, samples were restricted to persons living in non-institutionalized households, thus excluding residents of military barracks and dormitories. The generalisability of our findings to these excluded settings may therefore not be valid. Finally, the time lag between data collection and documentation of these findings is significant with the possibility that current realities may be different.

Despite these limitations, this study is strong enough to provide useful information about the use of SLT products in Nigeria and would be important to guide future research on SLT consumption. It would also help health policy-makers in planning tobacco control measures in Nigeria.

## Conclusions

Our study provides information about prevalence and patterns of SLT use among non-institutionalised adults in Nigeria and confirms that SLT use was higher among those with low-risk perception of SLT use, men, the older adults, rural residents, those with no formal education and residents of the north- and south-eastern regions of the country. A significant and comparatively high proportion of the population was not aware of the harmful effects of SLT use. Policymakers should consider the social distribution of use to provide context-specific SLT prevention and control strategies. There should be an emphasis on intense educational and enlightenment campaigns on the risks involved in SLT use so as to reduce the proportion of those unaware of use risk. Policymakers need to consider SLT use separately in tobacco control efforts since the risk factors and health effects of SLT use are different from that of smoking. It is also noteworthy that a follow-up GATS survey in Nigeria is long overdue.

## Author Contributions

**Conceptualization:** Ikenna Onoh, Oluwatomi Owopetu, Abdulhakeem Abayomi Olorukooba, Chukwuma David Umeokonkwo, Tukur Dahiru, Muhammad Shakir Balogun.

**Data curation:** Ikenna Onoh.

**Formal analysis:** Ikenna Onoh.

**Methodology:** Ikenna Onoh, Oluwatomi Owopetu, Abdulhakeem Abayomi Olorukooba, Chukwuma David Umeokonkwo, Tukur Dahiru, Muhammad Shakir Balogun.

**Supervision:** Ikenna Onoh, Abdulhakeem Abayomi Olorukooba, Chukwuma David Umeokonkwo, Tukur Dahiru, Muhammad Shakir Balogun.

**Validation:** Ikenna Onoh.

**Writing – original draft:** Ikenna Onoh, Oluwatomi Owopetu.

**Writing – review & editing:** Ikenna Onoh, Oluwatomi Owopetu, Abdulhakeem Abayomi Olorukooba, Chukwuma David Umeokonkwo, Tukur Dahiru, Muhammad Shakir Balogun.

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
