## [Decision Letter · Decision Letter 0]

16 Nov 2020

PONE-D-20-25941

Prevalence, patterns and risk factors of smokeless tobacco use in Nigerian adults: An analysis of the Global Adult Tobacco Survey

PLOS ONE

Dear Dr. Onoh,

Thank you for submitting your manuscript to PLOS ONE. After careful consideration, we feel that it has merit but does not fully meet PLOS ONE’s publication criteria as it currently stands. Therefore, we invite you to submit a revised version of the manuscript that addresses the points raised during the review process.

We look forward to receiving your revised manuscript.

Kind regards,

Stanton A. Glantz

Academic Editor

PLOS ONE

Journal Requirements:

2. Please state whether you accounted for regional clustering of the data in your statistical analyses. Did you consider using multilevel models. If not, please provide the rationale for not doing so.

Reviewers' comments:

Reviewer's Responses to Questions

**Comments to the Author**

1. Is the manuscript technically sound, and do the data support the conclusions?

Reviewer #1: Yes

Reviewer #2: Yes

2. Has the statistical analysis been performed appropriately and rigorously? 

Reviewer #1: Yes

Reviewer #2: Yes

3. Have the authors made all data underlying the findings in their manuscript fully available?

Reviewer #1: Yes

Reviewer #2: Yes

4. Is the manuscript presented in an intelligible fashion and written in standard English?

Reviewer #1: Yes

Reviewer #2: Yes

5. Review Comments to the Author

Reviewer #1: I appreciate author for selecting this topic for research. The article is nicely written however following are the few comments that need to be addressed

Sampling details in methodology need to be mentioned briefly. Although author has cited the final report yet brief details of household and participant selection may be provided in methods section.

Details regarding ethical review committee clearance or statement regarding work carried out as per Helsinki declaration need to be mentioned in text for the survey (from which data has been used for secondary analysis)

Reviewer #2: PONE-D-20-25941

The manuscript is a secondary analysis of 2012 GATS data in which the authors examine SLT use among Nigerian adults. The authors report that current SLT use was more common for those living in specific parts of the country, for those who were middle-aged, and for those who had lower harm perceptions. The paper is generally well-written. The topic is potentially important and results could lay the groundwork for future research that would increase understanding of and seek to address SLT use among Nigerians.

Comments

1. The abstract should probably include at least a brief indication of the rationale of the study – ie, the extent to which SLT use among Nigerian adults is a problem.

2. The title refers to prevalence, patterns, and risk factors, but I think something more like “prevalence and correlates” would be more accurate given the cross-sectional nature of the data. This arises again at the end of the intro.

3. On lines 54-55 the authors make the point that less attention has been paid to the “similarly harmful” SLT. It may be worth adding the difference in prevalence here, which is likely the reason for the difference in attention.

4. On line 68 a modifier is needed – is this prevalence of current use? And how is that operationalized? This is true of the rates in the next paragraph as well.

5. In the methods section, given the non-experimental methodology, the labels “dependent variable” and “independent variables” are not correct. There is no IV manipulation. These should be outcome and predictor variables or something similar.

6. Because there were no hypotheses I wonder whether the alpha level should be adjusted? Given the sample size it is unlikely to make much difference.

7. Suggest re-writing line 172 a bit. It could be read to suggest that very few were daily users (1.4% of 1.9% would be roughly 0.3%) but I believe this is 1.4% of the total, ie most who were current users were daily users.

8. Lines 182 and 198, recommend rewording “ignorant”.

9. I appreciate the way that the authors interpreted their ORs. I’m not sure there is value in both Tables 2 and 3 though and would suggest eliminating the former.

10. Is there any value in looking for differences by route of administration?

11. Line 245 refers to “drinking tobacco”; it is unclear what this means.

12. What do the authors make of the apparent contradiction between low levels of awareness of SLT harms and low prevalence?

6. PLOS authors have the option to publish the peer review history of their article (what does this mean?). If published, this will include your full peer review and any attached files.

Reviewer #1: **Yes: **Ibrar Rafique

Reviewer #2: No

---

## [Author Response · Author response to Decision Letter 0]

15 Dec 2020

Nigeria Field Epidemiology and Laboratory Training Programme

15th December, 2020

Dear Dr Glantz,

Response to Reviewers: PONE-D-20-25941

Thank you for the careful review of our manuscript. After careful consideration of the reviews, I hereby write this rebuttal letter to respond to the points and comments raised by the reviewers.

The following are the responses to the points raised by the reviewers:

A. Response to Academic Editor:

1. Comment:

Please state whether you accounted for regional clustering of the data in your statistical analyses. Did you consider using multilevel models? If not, please provide the rationale for not doing so.

Response: 

In the first part of our analyses (descriptive and bivariate), we did not need to account for clustering, but for unequal sampling probabilities of the complex survey design. We used a design-based approach to specify the complex survey design variables such as sampling weight, stratum and cluster to ensure that the unequal sampling probabilities are adjusted for thus yielding valid population parameters.

Before commenting on the second part of our analyses, it would be necessary to clearly understand the unique sample design features of the Nigerian 2012 GATS. The entire country was first stratified by its 36 states and the Federal Capital Territory within 6 regions. Within each state and by extension region, a sample of enumeration areas (EAs) were systematically selected from a master sample, with probability proportional size (PPS). These EAs constituted the primary sampling units (PSU) for the survey. A systematic sample of households (secondary sampling units, SSU) were selected from each EA, and then an eligible individual from each selected household. These EAs were mostly arbitrary creations with their aim being to aid operationalisation of the 2006 census. Even though they may correspond sometimes, they are not strict natural subdivisions with shared social commonalities.

The second part of our analyses, the regression modelling, is where accounting for the effect of clustering on the relationships between multiple levels of variables and SLT use with the aid of multilevel modelling was desired. However, there are a few points to note about multilevel modelling. Multilevel models (MLMs) facilitate inferences regarding variability among randomly selected clusters (e.g. EAs) in outcomes of interest and identify covariates that explain the variance in a given outcome at each level of a particular data hierarchy. The key thing that defines a variable as being a level (EAs and households/individuals) is that its units can be regarded as a random sample from a wider population of units, but this does not preclude the concept of natural clustering at higher divisions (states and regions) not accounted for in the sample design. 

The primary difficulty with the use of multilevel modelling in complex survey data revolves around incorporating sampling weights into the models. Failure to account for the unequal selection probabilities using sampling weights in MLMs can lead to biased parameter estimates. Including the weights as raw weights instead of correctly ‘scaled’ weights results in biased parameters and standard errors. A major practical obstacle in pseudo-maximum-likelihood (the underlying estimation approach for MLM that accounts for stratification, clustering and weighting) estimation for multilevel modelling of complex survey data is that most publicly available data do not include weights for each level of analysis. Most of them include only a single overall weighting variable for the lowest level units, whereas the pseudo-maximum-likelihood approach requires the weights corresponding to the levels of the hierarchical sampling design. If the sampling weights are ignored at either level, the parameter estimates can be substantially biased.

In as much as MLM was desired and considered, we were unable to employ this approach for a few reasons. Sample weights were only available at the individual level. Due to the nature and origin of the EA (the clustering variable by design), we considered it implausible as an important source of dependency in SLT use. There were also no cluster or higher-level variables available for building an MLM.

We therefore did not account for regional clustering as this could only be done using MLMs. Even if we were able to build an MLM, it would have been difficult to incorporate region as a level due to the survey’s particular sampling design.

2. Comment:

Please amend either the title on the online submission form (via Edit Submission) or the title in the manuscript so that they are identical.

Response: Effected by amending the title in the manuscript

B. Response to Reviewer #1:

1. Comment:

Sampling details in methodology need to be mentioned briefly. Although author has cited the final report yet brief details of household and participant selection may be provided in methods section.

Response: This has been done by adding brief details of stages of sampling.

2. Comment:

Details regarding ethical review committee clearance or statement regarding work carried out as per Helsinki declaration need to be mentioned in text for the survey (from which data has been used for secondary analysis)

Response: Further information on ethical considerations as available in the survey report has been added.

C. Response to Reviewer #2:

1. Comment:

The abstract should probably include at least a brief indication of the rationale of the study – ie, the extent to which SLT use among Nigerian adults is a problem.

Response: Minimal work has been done in this regard, and this was the major reason for this study. The 3rd sentence in the introduction to the abstract has been modified to reflect this.

2. Comment:

The title refers to prevalence, patterns, and risk factors, but I think something more like “prevalence and correlates” would be more accurate given the cross-sectional nature of the data. This arises again at the end of the intro.

Response: This has been noted and correction effected in the title, abstract and introduction.

3. Comment:

On lines 54-55 the authors make the point that less attention has been paid to the “similarly harmful” SLT. It may be worth adding the difference in prevalence here, which is likely the reason for the difference in attention.

Response: This has been added in the preceding sentence

4. Comment:

On line 68 a modifier is needed – is this prevalence of current use? And how is that operationalized? This is true of the rates in the next paragraph as well.

Response: It has been indicated that this prevalence speaks about current use as all quoted prevalences. The operationalization is as obtained in all GATS surveys: the numerator is number of current daily and less than daily smokeless tobacco users while the denominator is total number of respondents. The operationalization for the Nigeria Demographic and Health Survey (NDHS) is similar. 

5. Comment:

In the methods section, given the non-experimental methodology, the labels “dependent variable” and “independent variables” are not correct. There is no IV manipulation. These should be outcome and predictor variables or something similar.

Response: “Dependent” has been changed to “Outcome”, but we have decided to retain the use of “independent” as we believe the term “predictor” is more suggestive of causation or dependence.

6. Comment:

Because there were no hypotheses I wonder whether the alpha level should be adjusted? Given the sample size it is unlikely to make much difference.

Response: We have to chosen to retain the alpha level at 5%.

7. Comment:

Suggest re-writing line 172 a bit. It could be read to suggest that very few were daily users (1.4% of 1.9% would be roughly 0.3%) but I believe this is 1.4% of the total, ie most who were current users were daily users.

Response: It has been re-written to read “About three-quarters of this 1.9%, i.e., 1.4% (95% CI: 1.1%-1.8%) of the total adult population were daily users…”

8. Comment:

Lines 182 and 198, recommend rewording “ignorant”.

Response: This has been reworded to read “unaware”

9. Comment:

I appreciate the way that the authors interpreted their ORs. I’m not sure there is value in both Tables 2 and 3 though and would suggest eliminating the former.

Response: We appreciate this suggestion, but we have retained both tables as the contrast between the two demonstrates the effect of “adjustment” on the relationships between the factors and SLT use.

10. Comment:

Is there any value in looking for differences by route of administration?

Response: We do not see any extra utility in exploring differences by route of administration. In addition, due to the small numbers and percentages involved, these sub-group analyses may yield a lot of invalid results.

11. Comment:

Line 245 refers to “drinking tobacco”; it is unclear what this means.

Response: Smoking tobacco is a relatively uncommon form of use that involves boiling tobacco in water and drinking the water afterwards.

12. Comment:

What do the authors make of the apparent contradiction between low levels of awareness of SLT harms and low prevalence?

Response: The use of the term “low levels of awareness of SLT harm” is relative to levels for manufactured cigarettes and other countries. However, as much as 68.9% of Nigerian adults, a clear majority, are aware of SLT harm. Note that awareness of harm is not the only factor driving levels of use. There are other critical drivers such as cultural and social norms, and acceptability of the habit. The key thing about awareness levels and use status from our study is that there is a clear association between lack of awareness of harm and likelihood of use.

Yours sincerely, 

Ikenna Onoh MBBS, MSc

Resident, Nigeria Field Epidemiology and Laboratory Training Programme

ikeonoh@yahoo.com; +2347036354733

---

## [Editor Report · Decision Letter 1]

17 Dec 2020

PONE-D-20-25941R1

Prevalence, patterns and correlates of smokeless tobacco use in Nigerian adults: An analysis of the Global Adult Tobacco Survey

PLOS ONE

Dear Dr. Onoh,

Thank you for submitting your manuscript to PLOS ONE. After careful consideration, we feel that it has merit but does not fully meet PLOS ONE’s publication criteria as it currently stands. Therefore, we invite you to submit a revised version of the manuscript that addresses the points raised during the review process.

In your abstract you say, “However, the perception of possible harm from its use was low. We recommended targeted interventions to increase awareness of the harmful effects.”  Your response to Reviewer 1 had a more specific and informative statement:  “The use of the term “low levels of awareness of SLT harm” is relative to levels for manufactured cigarettes and other countries. However, as much as 68.9% of Nigerian adults, a clear majority, are aware of SLT harm. Note that awareness of harm is not the only factor driving levels of use. There are other critical drivers such as cultural and social norms, and acceptability of the habit. The key thing about awareness levels and use status from our study is that there is a clear association between lack of awareness of harm and likelihood of use.”

Please revise the abstract, discussion and conclusion of the paper to reflect this more sophisticated explanation.  Please also discuss the “other factors” driving use and how consideration of those factors could inform prevention efforts.

We look forward to receiving your revised manuscript.

Kind regards,

Stanton A. Glantz

Academic Editor

PLOS ONE

---

## [Author Response · Author response to Decision Letter 1]

21 Dec 2020

Nigeria Field Epidemiology and Laboratory Training Programme

22nd December, 2020

Dear Dr Glantz,

Response to Reviewers: PONE-D-20-25941

Thank you for the careful review of our manuscript. After careful consideration of the reviews, I hereby write this rebuttal letter to respond to the points and comments raised by the reviewers.

The following are the responses to the points you raised:

A. Response to Academic Editor:

1. Comment:

In your abstract you say, “However, the perception of possible harm from its use was low. We recommended targeted interventions to increase awareness of the harmful effects.” Your response to Reviewer 1 had a more specific and informative statement: “The use of the term “low levels of awareness of SLT harm” is relative to levels for manufactured cigarettes and other countries. However, as much as 68.9% of Nigerian adults, a clear majority, are aware of SLT harm. Note that awareness of harm is not the only factor driving levels of use. There are other critical drivers such as cultural and social norms, and acceptability of the habit. The key thing about awareness levels and use status from our study is that there is a clear association between lack of awareness of harm and likelihood of use.”

Please revise the abstract, discussion and conclusion of the paper to reflect this more sophisticated explanation. Please also discuss the “other factors” driving use and how consideration of those factors could inform prevention efforts.

Response: 

Adjustments have been made to reflect this more sophisticated explanation as advised. These adjustments are in the concluding part of the abstract, the paragraph spanning lines 262 to 273 of the revised unmarked manuscript and in the conclusion.

Yours sincerely, 

Ikenna Onoh MBBS, MSc

Resident, Nigeria Field Epidemiology and Laboratory Training Programme

ikeonoh@yahoo.com; +2347036354733

---

## [Editor Report · Decision Letter 2]

23 Dec 2020

Prevalence, patterns and correlates of smokeless tobacco use in Nigerian adults: An analysis of the Global Adult Tobacco Survey

PONE-D-20-25941R2

Dear Dr. Onoh,

We’re pleased to inform you that your manuscript has been judged scientifically suitable for publication and will be formally accepted for publication once it meets all outstanding technical requirements.

Kind regards,

Stanton A. Glantz

Academic Editor

PLOS ONE
---

## [Editor Report · Acceptance letter]

28 Dec 2020

PONE-D-20-25941R2 

Prevalence, patterns and correlates of smokeless tobacco use in Nigerian adults: An analysis of the Global Adult Tobacco Survey 

Dear Dr. Onoh:

I'm pleased to inform you that your manuscript has been deemed suitable for publication in PLOS ONE. Congratulations! Your manuscript is now with our production department. 

Kind regards, 

on behalf of

Professor Stanton A. Glantz 

Academic Editor

PLOS ONE